# Patellofemoral Pain, Q-Angle, and Performance in Female Chinese Collegiate Soccer Players

**DOI:** 10.3390/medicina59030589

**Published:** 2023-03-16

**Authors:** Songhui You, Yinhao Shen, Qingguang Liu, Antonio Cicchella

**Affiliations:** 1Siping Road Campus, International College of Football, Tongji University, Shanghai 200092, China; 2Department for Quality-of-Life Studies, University of Bologna, 47921 Rimini, Italy

**Keywords:** female football, knee pain, fatigue, performance, soccer, Q-angle, jumping, match time

## Abstract

*Background and objective:* Female sports injuries have been neglected by science, and few relevant studies have considered female subjects. Knee pain in female soccer players is more common than in male soccer players. The number of days of absence from training and competition has been shown to be higher in females than males. The reporting of knee pain is common in female soccer players, but whether knee pain is associated with morphological features is unclear. The Q-angle of the knee has been hypothesized to be a causal factor in knee pain. Asian females have shown higher levels of valgus than non-sporting Caucasian populations, but no data exist for female Chinese players. The aim of our study was to investigate whether there are associations between knee pain, the Q-angle of the lower limb, jump performance, play time, and perceived exertion in female Chinese collegiate soccer players. *Materials and Methods:* We measured the Q-angle, patellofemoral/anterior knee pain (SNAPPS questionnaire), and CMJ and SJ performance of 21 subjects (age: 20.09 ± 1.13 years, weight: 56.9 ± 6.26 kg, height: 164.24 ± 4.48 cm, and >10 years of practice) before and after a match; Borg scale and play time results were also recorded. *Results:* We found that our studied group had higher Q-angles in comparison to other ethnic groups reported in the literature, as well as an association of the Q-angle with the age, height, and weight of the players; however, contrary to other studies, we did not find any association between the Q-angle and knee pain, jumps, play time, or perceived exertion. Knee pain was not associated with any of the measured variables. *Conclusions*: Female Chinese soccer players showed higher Q-angles than players of other ethnic groups, a result that was associated with anthropometrics. The Q-angle was not found to be associated with knee pain, for which the sole determinant was body height.

## 1. Introduction

Our study was motivated by the high incidence of knee pain in the female collegiate team of the International College of Football at Tongji University in Shanghai. We thus aimed at searching for (if any) associations that exist in our sample between knee pain, jumps, perceived fatigue, playing time, and Q-angle. A recent review reported that only 9% of published studies on sports injuries exclusively focused on female players, despite their higher observed incidence of knee injuries [1]. The injury rate in females is 27.6 injury incidences/1000 h exposure in international-level female football [2], and females face more consequences than males (506.7 vs. 454.0 days of absence per 1000 h, respectively) [2]. The burden of knee pain in female soccer players begins in adolescence following growth spurts [3,4] and is probably linked to the faster growth of bone compared with soft tissues and muscle. A previous study showed that 50% of retired female soccer players with a history of knee injuries develop osteoarthritis before the age of 50 (mean age: 37 years) [5]. The prevalence of valgus knee (which leads to incorrect biomechanics when landing from jumps) has been hypothesized as one reason for the higher incidence of injuries in female soccer players compared with male players [6]. It has been shown that when landing from a drop jump, females land in a more valgus knee position and have poorer knee control than men [7]. Basic anatomical findings in cadaveric limbs showed that female knees have greater valgus laxity, with 0° to 50° of flexion, compared with male knees [8].

In clinical practice, the presence of a valgus knee is determined via the quadriceps angle (Q-angle) using different methods: statics, dynamics, or radiographic/MRI techniques [9,10,11,12,13,14,15]. Usually, radiographic techniques are used to consider tibial rotation, but alternative and non-irradiating methods are preferable due to the diffusion of surface measurements. The Q-angle is defined as the angle between a line drawn from the spina iliaca anterior superior and the center of the patella and a line drawn from the center of the patella to the center of the tibial tubercle. Q-angle measurement is hypothesized to be relevant to not only orthopedic surgery and physical therapy but also functional performance in sport. Consensus about reference values indicates that a Q-angle exceeding 15 degrees in males and 20 degrees in females is clinically “abnormal” [3], at least in Caucasians. In fact, the Q-angles reported in the literature vary with ethnicity [9,10,11,12,13,14,15], e.g., from 26.54 ± 7.01 degrees in Turkish volleyball players to 13.7 ± 0.49 degrees in Arabian women. Soccer, volleyball, and basketball were shown to be the sports with the highest incidence of patellofemoral pain (PFP) injuries in collegiate sports in the USA [16].

The contraction of quadriceps muscles straightens the Q-angle, so sports using a high amount of quadriceps (strength) training are associated with lower Q-angles because of the pulling effect of the quadriceps muscle [17]. However, pulling dynamics around the knee joint comprise a complex factor due to anatomical structure interactions [17]. The reliability of the Q-angle has been questioned [18] because of methodological problems such as the lack of a standardized measurement procedure and the lack of proper assessor training [19]. Despite these critiques, it remains a widely used clinical tool for the assessment of valgus knee and relevant research. The Q-angle has been related to several clinical conditions such as chondromalacia patellae, patellofemoral pain (PFP), and ACL injuries [20,21,22], but does not seem to be a causative factor for ankle sprains [23]. It must be noted that patellofemoral pain and anterior knee pain are often used interchangeably in the literature. Furthermore, higher Q-angles were shown to be associated with performance, specifically with decreased isokinetic knee strength, power output, and torque [24]. Another study on Chinese females (n.676, 18–40 years old) reported PFP in 21.2% of their sample [25,26] and did not demonstrate any associations with BMI, age, or gender. This study used the same questionnaire employed in this study (described in the Methods section). Greater Q-angles (valgus) were previously observed in Asian females compared with Caucasian females [26]. Furthermore, another study observed a significant difference between male and female Japanese subjects, with women exhibiting a higher percentage of valgus alignment (50%) than men (36%) [27]. Significant differences in knee morphology between Chinese and Caucasian cohorts, which resulted in the greater valgus alignment of the distal femurs of Chinese women, were also noticed. This difference can also explain the higher incidence of mono-compartmental lateral osteoarthritis in Asian women [26].

It was hypothesized that the Q-angle influences the biomechanics of the lower limbs [18,24] and thus the basic performance of functions such as jumping [6]. It was also found that there were relationships between static and dynamic Q-angles (e.g., during walking, jumping, and other functional movements) [28]. The association of the Q-angle with vertical jumps is controversial in the literature, though little research has been conducted on the topic. In mixed groups (male and female), two studies found a negative correlation between SJ (squat jump, performed from a fixed knee angle of 90 degrees, with hands on the hips) [29] and CMJ (counter-movement jump, from the standing starting position followed by knee flexion and extension, with hands on the hips) [30] performance and the Q-angle, while another study found no correlation in university students [31]. Specifically, jumping height was found to be associated with a smaller Q-angle [29]. A study of female collegiate varsity athletes did not find an association between the vertical jump and the Q-angle [32]. However, the Q-angle was found to be associated with height in females [33], and height was found to be associated with playing time [34]. Furthermore, vertical jump measurements can provide valuable insight into the neuromuscular qualities of female soccer athletes. A recent study found a decrease of 6.1% in CMJ performance in female soccer players after a match [35]. Playing time is a general index of performance; a player can be kept on the field for an entire game or be dismissed due to poor performance. Play time has therefore been assumed to be a gross indicator of player quality [34]. A previous study showed an increase in Borg’s perceived exertion of +25% [36] after a match. It has been suggested that studies considering half-match duration (45 min) can be used to better understand the performance/fatigue relationships of female soccer players [36]. Knee pain has been investigated in soccer and other sports, and the results clearly show that soccer is the most at-risk sport [37]. Biomechanical comparisons of painful and asymptomatic knees during running, a basic soccer task, were performed [38], and it was found that the frontal plane projection angle was significantly larger in painful knees than in asymptomatic knees during a single leg squat. Questionnaires remain the only way to measure the theoretical construct of pain, and in the literature, several questionnaires are available [39,40,41,42]. A study on female basketball players employed the VISA-P questionnaire [39], reporting the incidence of anterior knee pain in 34% of 64 young (11–15-year-old) female basketball players [4]. Other available questionnaires are the KOOS, which refers to osteoarthritis [40]; the Knee-Cap Questionnaire (KC) [41], which has a section on affective and sensory perceptions; and the Oxford Knee Score (OKS), which is more suitable for assessing the pain and function of patients undergoing knee replacement surgery [42]. Another available questionnaire for PFP/anterior knee pain is the SNAPPS, which has been used in large samples of normal populations [24,43]. However, there have been no cross-validation studies on the KOOS, KC, VISA-P, OKS, and SNAPPS questionnaires. The SNAPPS questionnaire was validated for the Chinese language and was previously used in a large study on a Chinese population, which served as a reference for normal data in our study [25].

The aim of this study was to investigate if there are associations between knee pain, jumps, perceived fatigue, playing time, and Q-angle in a sample of female semi-professional Chinese collegiate soccer players.

## 2. Methods

Participants. In this prospective transversal study, a total of 21 female collegiate soccer athletes from the International College of Football at Tongji University in Shanghai who play in the Chinese intercollegiate league were enrolled in this study. The inclusion criteria were to be a female member of the Tongji University collegiate football club. The players trained every day for at least 2 h and participated in the National Chinese Collegiate League championship, having at least one match per week. All the players had >10 years of practice, had no knee surgery, and were in generally good health at the time of testing. Health status was assessed by a clinician. After the explanation of the research aims, all participants were of Han ethnicity. Informed consent was obtained from the athletes. All players were residential students on the university campus and were recruited via direct contact with coaches. The study was conducted according to the guidelines of the Declaration of Helsinki and approved by the Ethics Committee of Tongji University (approval code: tjdxsr029).

### Procedures

Pain scores were calculated using the Chinese version of the SNAPPS questionnaire developed by Dey et al. [44], using Section 2 and Section 4 as recommended. SNAPPS stands for Survey Instrument for Natural History, Aetiology, and Prevalence of Patellofemoral Pain Studies. It is a screening questionnaire used to identify people with patellofemoral pain in populations between the ages of 18 and 40, and it was previously used in a Chinese population [25].

The questionnaire is available online [45]. The questionnaire was filled out at rest, not close to training sessions or matches. According to SNAPPS, a score of 6 or above indicates knee pain. Q-angle measurements were performed according to the method proposed by Merchant et al. in the supine rest position with relaxed quadriceps and joined legs.

To identify the patella borders before the measurements, subjects were asked to contract their quadriceps. After the identification of the borders of the patella, the tibial tubercle, and the ASIS, black dots were assigned to guide the alignment of a goniometer. All measurements were performed twice per leg by the same trained assessor, and the mean value was retained. The participants were positioned in the dorsal decubitus position with the knee and hip extended and the hip and foot in neutral rotation using an anthropometric goniometer [46] for both legs. Jumps were measured before and after a collegiate match (within 1 h) using the My Jump ver. 2.0 app [47]. The subjects performed three maximal trials, and the best jump was retained. The muscle elasticity index was calculated from the height reached during the CMJ and SJ (CMJ-SJ * 100/SJ) [48,49]. Perceived exertion was measured within 1 h after a match with the Borg CR-10 scale in Mandari [50], and the match time for each player was also recorded.

Statistical analysis. Correlation, paired sample t-tests, and linear regression were performed in SPSS v. 25.0. with a confidence interval of 95%. All variables were normally distributed according to the Kolmogorov–Smirnoff test. The sequence of measurements is depicted in Figure 1.

## 3. Results

The subjects had an age of 20.09 ± 1.13 years, a weight of 56.9 ± 6.26 kg, a height of 164.24 ± 4.48 cm, and a BMI of 19.89 ± 1.55.

Descriptive statistics are reported in Table 1.

The mean match time was similar to the half-time of a football match and allowed for inferences about submaximal effort, as suggested by a previous study [36].

The Q-angles showed negative correlations with age (r = −0.485, *p* = 0.026), weight (r = −0.538, *p* = 0.012), and height (r = −0.564, *p* = 0.008) (Figure 1).

No significant differences were found between the painful and asymptomatic subjects in all measured variables. Among all players, six subjects reported a pain score of 6 or above (28.5% of the sample), which is the threshold score in the SNAPPS questionnaire for knee pain. This percent is close to what was found in a previous study on young, female, non-sporting Chinese people (21.2%) [25]. Pain scores were not correlated with any measured variables in a previous study on non-sporting Chinese females [23], according to experts’ consensus [51], or in other populations [52]. Here, no significant differences were found between the left and right Q-angles. In our sample, the Q-angle values were above the means found in the literature for similar populations (Table 1). We did not find any correlation between Q-angle with jumps and elasticity index, perceived fatigue, and match time. Height was not correlated with match time. Performances by CMJ and SJ significantly decreased after a match. SJ performance decreased by 11.3% (t = 2.363, *p* = 0.028, Cohen’s D = 0.24) and CMJ performance decreased by 4.7% (t = 3.333, *p* = 0.003, Cohen’s D = 0.33) after a 44-min match (mean), though the elasticity index did not significantly change after such a match. The after-match Borg CR-10 score (Figure 2) was found to be 5 ± 3 (strong effort) and was highly correlated with playing time (r = 853, *p* = 0.000). No predictor of pain score included the Q-angle, which was not associated with PFP/knee pain in contrast to previous studies on non-sporting Asian [10] and Caucasian [32,53] females.

## 4. Discussion

Among the objectives for the 2022–2035 strategic plan of the Chinese Ministry of Sport is a push to further develop women’s Chinese football [54]. Therefore, there is a need for scientific information to guide the training process for female athletes and the rehabilitation process following injuries, specifically in the Chinese population. Knee pain is a condition that alters the training process and must be considered by coaches. Unfortunately, female injuries in sport have been under-considered by scientists for many years, so coaches’ awareness of them is also low [1]. Knee pain is often seen as an inevitable condition and is accepted as a side effect of the training process by both coaches and athletes. The long-term consequences of knee pain [5] are neglected. After long years of training and playing, knee pain can become a disabling condition that leads to osteoarthritis or more severe conditions, impairing gait and daily life activities [5]. The health costs associated with injuries in athletes have social and individual impacts [55]. For instance, footballers can produce substantial healthcare and societal costs, which has important implications for healthcare provisions as well as the prioritization and implementation of injury prevention programs and post-injury rehabilitation. The aims of this study were to investigate a neglected aspect of soccer injuries and explore a possible association between some anthropometrics, performance variables, and pain in a cohort of highly trained and high-performing female players. Our findings on female Chinese collegiate soccer athletes showed the Q-angle to be above the mean for different ethnicities and sports reported in the existing literature [10,16]. This result could reflect a characteristic of the female Chinese (Asian) knee, which presents an accentuated valgus compared with females of Caucasian and other ethnicities, as previously observed in non-sporting female Chinese populations [24,26]. A higher level of knee valgus has been reported in female Asian soccer players compared with males, as has also been found in Caucasians, and this finding has been associated with differences in knee pain between males and females [56]. However, the results of our study do not support these findings [9]. In contrast to a study on Arabian females [33], we found a significant correlation between the Q-angle and body weight, which can also be explained by the different morphologies of Asian somatotypes. It is worth noting that the study on Arabian females reported increases in the Q-angle with body weight, though the statistical association did not reach significance. Body weight, on the other hand, is correlated with muscle mass, as heavier subjects have more lower limb muscle mass and, therefore, stronger pulling effects on the Q-angle. The association of the Q-angle with age may be explained by the fact that older athletes have stronger quadriceps muscles, which straighten the Q-angle, as previously hypothesized [18]. We found that knee pain in female Chinese collegiate soccer players was not associated with the Q-angle, contrary to what was observed in non-sport Asian women and other populations [8,11]. In our sample, which was highly homogeneous, knee pain was present in 28.5% of the sample, and no differences were found between painful and asymptomatic subjects in all measured variables. These results are consistent with previous findings on male runners, where the dynamic Q-angle did not change with pain [44], but there has been no similar study on female runners. We did not find height to be a predictor of pain score, which disagrees with the results of a previous study on adolescent male basketball players [57] that showed that the knees of taller subjects must endure more mechanical stress than those of shorter subjects due to the increased torque acting on the knees’ structures [57]. However, this idea did not seem to apply to our sample of female Asian soccer players. Our results lead to questions regarding the usefulness of Q-angle measurements for predicting knee pain in heavily trained Chinese (Asian) collegiate soccer players, in accordance with a previous study [19] that questioned the usefulness of the Q-angle as a predictor of pain. These results have clinical and practical relevance. First, it is not necessary to try to reduce the Q-angle by means of strength training; second, it does not seem useful to use this measure for female Chinese soccer players. Previous studies found a correlation between the Q-angle and isokinetic strength in sport students [24] and isometric strength in a female population [58]. High-speed (240 degrees/sec) isokinetic strength was found to be correlated with jumps in young elite female basketball players [59]. We did not find any correlations between the Q-angle and jump performance, and though there are currently no relevant data in the literature regarding female soccer players, the same result was previously obtained for female varsity athletes [30]. In addition to our major findings, we did not find any relationship between playing time and athlete height. This result is in accordance with a previous study [60] and suggests questions regarding the NCAA guidelines for recruiting female soccer players, which recommend recruiting females taller than 165 cm due to an association between height and playtime [60]. To our knowledge, the present study was the first to investigate female Chinese soccer players, and our results can be used as a reference and a guide to better understand how to design training to avoid knee pain, which has been reported by high numbers of players. The results of our study also raise awareness regarding the high prevalence of knee pain in female Chinese collegiate soccer players. A limit of our study is the measurement of the Q-angle itself. Although it is a widely used measure, some doubts remain regarding its reliability, and there is no gold standard procedure for its measurement. However, the Q-angle measurement method we chose has been shown to be reliable, validated in the Chinese language, and previously used in a large study on a Chinese population [61]. Other limitations of our study are that the definition of knee pain is vague and that the identification of painful points on one’s own knee can be problematic due to the diffuse nature of knee pain. However, there is no other method used to measure pain perception other than questionnaires. Psychological variables can be used to clarify the way that pain perception influences performance. However, only a few studies have recently investigated the psychological aspects of PFP and knee pain. Anxiety, depression, catastrophizing, and fear of movement may be elevated in individuals with PFP and be correlated with pain and reduced physical function [62]. These psychological factors could have altered the response to the SNAPPS questionnaire in our sample and require further investigation. Psychological factors also influence perceived exertion. Our sample showed a score range from 3 (moderate, non-playing, only warming up) to 9, with a mean score of 5.09 on the Borg CR-10 scale (strong). Adding a measurement of mood state could help to improve the understanding of these results. Some practical recommendations for coaches can be inferred from these results. Female soccer players’ knees are at-risk sites for injuries, and knee morphology (e.g., valgus) does not appear to be a determinant of pain; therefore, preventive exercises, such as the strengthening of quadriceps muscles, should be included in training routines.

## 5. Conclusions

In recent years, there has been greater awareness of the health and wellbeing of athletes beyond performance [63]. In summary, the results of our study (and those of previous studies) suggest that soccer is a risky activity for both males and females, but more so for females, and there are probably specific adverse effects associated with participation in professional football that are both immediate [64] and delayed [5].

The main results of our study are that the Q-angle was not found to be associated with pain score and that knee pain in female Chinese collegiate soccer players was found to be associated with age and anthropometric morphology, specifically height and weight. Our study showed that the Q-angles of female Chinese collegiate soccer players were higher than those in other ethnic groups. Further studies are needed to clarify the Q-angle/knee PFP pain relationships. Whether other causal factors intervene in knee pain requires further investigation. The results of our study also suggest the necessity of fostering research on female sports, especially female soccer players, due to the near-future development of female soccer in China.

## Data Availability

The data presented in this study are available on request from the corresponding author. The data are not publicly available due to privacy.

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
