# Peer review of "Patellofemoral Pain, Q-Angle, and Performance in Female Chinese Collegiate Soccer Players"

_medicina, 2023, doi:10.3390/medicina59030589_

Round 1

Reviewer 1 Report

Dear authors, first of all thank you for your submission to Medicina. The theme presented by the authors is of potential importance in view of the scarcity of research in this regard and therefore deserves the utmost consideration. Still, there are some points that need to be clarified and others tweaked to ensure that the work can reach its full potential.

Below you can find my comments in detail.

Although the introduction provides an adequate framework for the topic of study, the introduction is too long and there are some doubts about the need to include an explanatory table at this stage of the article. In my opinion this information should be diluted in the text. In addition, the information could be more synthesized, more assertive in order to facilitate reading.

The point "methods" appears all in a single paragraph. subdivide into topics to separate the information: (Ex: participants, procedures, statistical analysis, etc..)

I would like to see clarification on how the sample was calculated (power, effect etc..).

Table 2 presents the results, why are the P values not presented?

In the end, it seems fundamental to provide some practical recommendations for coaches in the light of the conclusions found.

Author Response

I resubmit, according to your observation.

Thank you very much for your help.

Antonio Cicchella

Reviewer 2 Report

Introduction

Line 59 – what is PFP?

The introduction was somewhat meandering in nature. It was difficult to follow the argument or identification of a research problem and a justification for the study by the authors.

I suggest the authors revise this section extensively. The problem identified should be lucid, unambiguous, and very clearly spelled out. this should be followed by a very definite research question, highlighting the goals of the study. this will form the basis for assessing the suitability of the methodology chosen, the validity of the results obtained, and the relevance of the conclusions reached.

I also suggest the authors use language editing software as it was sometimes difficult to follow the flow of thought based on grammatical errors.

Methodology

The methodology should focus on details of the planning of the study and how the evaluation was carried out. The authors should be pedantic enough to state whether this was a prospectively or retrospectively designed study. Details of the inclusion and exclusion criteria should be clearly spelled out.

Details such as mean age, weight, height, and BMI should be moved to the results section. furthermore, decimals are represented using a ‘period’ sign and not a ‘comma’ sign.

Q angle was measured with the quadriceps muscle in contraction (line 157)?

Author Response

I tried to address your suggestions. Thank you very much for the time spent on thie review.

Antonio Cicchella

Round 2

Reviewer 1 Report

I no longer have reservations about the manuscript after the changes made by the authors.

Author Response

Thank you very much for your precious suggestions.

Reviewer 2 Report

The manuscript is significantly improved though I still find the introduction some long and not straight to the point. 

 Yilmaz and co [ref 17], in their article, referred to the pull of the quadriceps muscle as their assumed reason for a reduction in the Q angle, and not the vastus medialis as the authors have stated on line 64. An isolated pull of the vastus medialis will more likely lead to an increase rather than a decrease in the Q angle. I encourage the authors to double check this statement and revise as necessary. 

The abbreviations SJ and CMJ appear to have been used for the first time on lines 94 and 95 respectively without clearly stating the full meaning of the abbreviations and their definitions.

Asides from these observations, I have no other issues with the manuscript. 

Sincerely yours

Author Response

Thank you very much, I enclosed a word file with the responses to yours suggestions.

Thank you again for the time spent reviewing our paper.
